# Ultra-sensitive and rapid detection of nucleic acids and microorganisms in body fluids using single-molecule tethering

Wen-Chih Cheng[1,3], Troy Horn[1,3], Maya Zayats[1], Georges Rizk[1], Samuel Major [1], Hongying Zhu[1], Joseph Russell[1], Zhiguang Xu [1], Richard E. Rothman[2] & Alfredo Celedon [1✉]

Detection of microbial nucleic acids in body fluids has become the preferred method for rapid diagnosis of many infectious diseases. However, culture-based diagnostics that are time-consuming remain the gold standard approach in certain cases, such as sepsis. New culture-free methods are urgently needed. Here, we describe Single MOLecule Tethering or SMOLT, an amplification-free and purification-free molecular assay that can detect microorganisms in body fluids with high sensitivity without the need of culturing. The signal of SMOLT is generated by the displacement of micron-size beads tethered by DNA probes that are between 1 and 7 microns long. The molecular extension of thousands of DNA probes is determined with sub-micron precision using a robust and rapid optical approach. We demonstrate that SMOLT can detect nucleic acids directly in blood, urine and sputum at sub-femtomolar concentrations, and microorganisms in blood at 1 CFU mL$^{-1}$ (colony forming unit per milliliter) threefold faster, with higher multiplexing capacity and with a more straight-forward protocol than amplified methodologies. SMOLT's clinical utility is further demonstrated by developing a multiplex assay for simultaneous detection of sepsis-causing *Candida* species directly in whole blood.

[1] Scanogen Inc., Baltimore, MD 21244, USA. [2] Department of Emergency Medicine, Johns Hopkins University, Baltimore, MD 21209, USA. [3]These authors contributed equally: Wen-Chih Cheng, Troy Horn. ✉email: aceledon@scanogen.com

The polymerase chain reaction (PCR) has enabled a revolution in in vitro diagnostics thanks to its sensitivity and specificity[1]. However, polymerase-based methodologies require complex sample preparation steps to remove polymerase inhibitors in certain specimen types, and relatively expensive reagents and instrumentation[2,3]. These limitations have complicated the development and implementation of PCR-based systems for some applications, such as culture-free sepsis diagnosis.

Sepsis is a life-threatening condition that can be caused by microorganisms in the bloodstream at concentrations as low as ~1 CFU mL$^{-1}$ [4,5]. Sepsis is associated with over 180,000 deaths in the USA annually[6], and a rapid (<3 h) and sensitive assay for this condition is urgently needed[7]. In spite of extensive research to advance molecular diagnostics methods for sepsis[7,8], only one PCR-based and culture-free system has been cleared by the Food and Drug Administration to diagnose this condition to date. That system has an average time-to-results of 4–8 h, with limits of detection (LOD) between 1 and 11 CFU mL$^{-1}$ depending on the species[9,10]. Many polymerase-free molecular detection methods have been developed, which could circumvent the limitations of PCR-based systems. Some examples include nanostructured microelectrode[11], bio-bar-code[12], single-molecule array[13], electrochemical biosensor[14], and surface plasmon resonance sensor[15]. However, many of these methods are time-consuming and have shown insufficient sensitivity for detection of microorganisms, with LODs between 10$^3$ and 10$^7$ CFU mL$^{-1}$ [11,13,14].

Here, we report SMOLT, a polymerase-free molecular assay capable of sensitive detection of microorganisms in blood with a turnaround time of 1.5 h. We show that SMOLT detects microbes directly in whole-blood samples with LODs between 1 and 3 CFU mL$^{-1}$.

## Results

**The SMOLT assay**. SMOLT detects nucleic acid targets using DNA-based Long Probes (>3 kb pairs) that tether micron-size beads inside a capillary when the target molecule is present (Fig. 1a). Beads tethered by a Long Probe can be discriminated from beads that non-specifically attach to the capillary based on the displacement they exhibit in the presence of liquid flow: Long-Probe tethered beads are displaced a significantly greater distance than non-specifically attached beads. The displacement of beads located in a large area (24 mm$^2$) is determined with sub-micrometer precision by processing images obtained with a low-magnification lens and a low-cost digital camera (See "Methods" and Supplementary Figs. 1, 2).

The SMOLT assay consists of three steps: lysis (4 min), hybridization (45 min) and detection (40 min) (Fig. 1b). During the lysis step, the body fluid is heated to 95 °C in the presence of a surfactant at high concentration (8% lithium dodecyl sulfate) to denature enzymes and lyse cells. This highly denaturing environment prevents aggregation and inactivates nucleases that otherwise would degrade target molecules and probes. We find that this simple step is sufficient to lyse the tested microorganisms without the need for bead beating nor enzymes. An aliquot from the lysate is then sequentially incubated with DNA probes and oligonucleotide-functionalized magnetic beads. The target molecule hybridizes to a single-stranded flap at one end of the Long Probe and to a DNA oligonucleotide, Oligo-2 (Fig. 1a). Oligo-2 then hybridizes to the oligonucleotide functionalized on the surface of the beads, Oligo-1. The formation of the Long Probe/target/bead complex is detected by introducing the sample into a capillary functionalized with Oligo-3, a DNA oligonucleotide complementary to the single-stranded flap on the other end of the Long Probe (see "Methods"). Inside the capillary, the Long Probe/target/bead complexes bind to Oligo-3. Unbound beads are then

removed from the capillary by washing buffer, and the beads that remain bound to the capillary surface are imaged in the presence of liquid flow to determine the distance they move. One capillary image is obtained with flow in one direction and another image is obtained when the flow direction is reversed (Fig. 1c). The position of each bead in the two images is determined as the centroid of the bead image area (see "Methods"). By comparing the position of each bead in the two images, the displacement of each bead is determined with high precision.

The displacement of all the beads in the field of view is used to determine the SMOLT signal. A histogram that summarizes bead displacement data is generated (orange line in Fig. 1d). The orange line in Fig. 1d shows the displacement of beads in an experiment where target molecules were present and includes beads that are the signal from the target but also beads that are noise. Signal beads are the ones tethered to the surface by the Long-Probe/target/bead complex (Fig. 1a) and, therefore, these beads move a specific distance. The peak at 5.5 μm in Fig. 1d (orange line) is mostly formed by signal beads. The orange line in Fig. 1e, obtained in the absence of target, exposes the presence of beads that are noise. These beads are non-specifically bound to the capillary surface or to the Long Probe without a target molecule. Non-specifically bound beads that are displaced a distance different from the distance that signal beads are displaced (<5 μm in Fig. 1d, e, orange line) are excluded from the SMOLT signal based on their displacement. However, there are non-specifically bound beads that are displaced the same distance as the signal beads (peak at 5.5 μm in Fig. 1e, orange line). We distinguish these non-specifically bound beads from signal beads using strand displacement to specifically disrupt the tether of the signal beads (Supplementary Fig. 3)[16]. We flow into the capillary strand-displacement oligonucleotides that are complementary to the target in the regions where the Long Probe and Oligo-2 hybridize to the target. Strand-displacement oligonucleotides dissociate the target from the probes by displacing the probes as they hybridize to the target. This process disconnects the signal beads from their tether, but it does not affect the tether of the noise beads. An alternative method to disrupt the tether of signal beads while not affecting the noise beads is to flow into the capillary RNase A. RNase A cleaves RNA molecules but it does not act on DNA molecules. Therefore, the enzyme disconnects the signal beads from their tether by cleaving the target, but it does not affect the tether of noise beads (Supplementary Fig. 3).

Discriminating beads based on bead displacement and disruption dramatically reduces background noise. We define the SMOLT signal as the number of beads that satisfy two conditions: (1) they move a specific distance when the flow in the capillary is reversed and (2) they detach from the capillary during the disruption step. The blue line in Fig. 1d, e is a histogram of bead displacement including only the beads that detached from the capillary during the disruption step. The SMOLT signal in Fig. 1d, e is the beads in the blue histogram between 5 and 5.8 μm. The background noise, indicated by the average number of beads detected in blank samples, is reduced from an average of 8870 beads throughout this study to 963 beads when the displacement requirement is applied, and to 12.5 beads when the displacement and disruption requirements are applied, indicating a 710-fold reduction in background noise (Supplementary Table 1).

**Detection of synthetic RNA directly in body fluids**. To study the performance of SMOLT in human body fluids, we evaluated the assay using synthetic RNA oligonucleotides spiked into whole blood, urine, and sputum. Each body fluid was spiked with target molecules and analyzed using the same SMOLT assay described above. Titration studies revealed that the assay limit of detection (LOD) was 0.11 fM in blood, 0.28 fM in urine, and 0.33 fM in

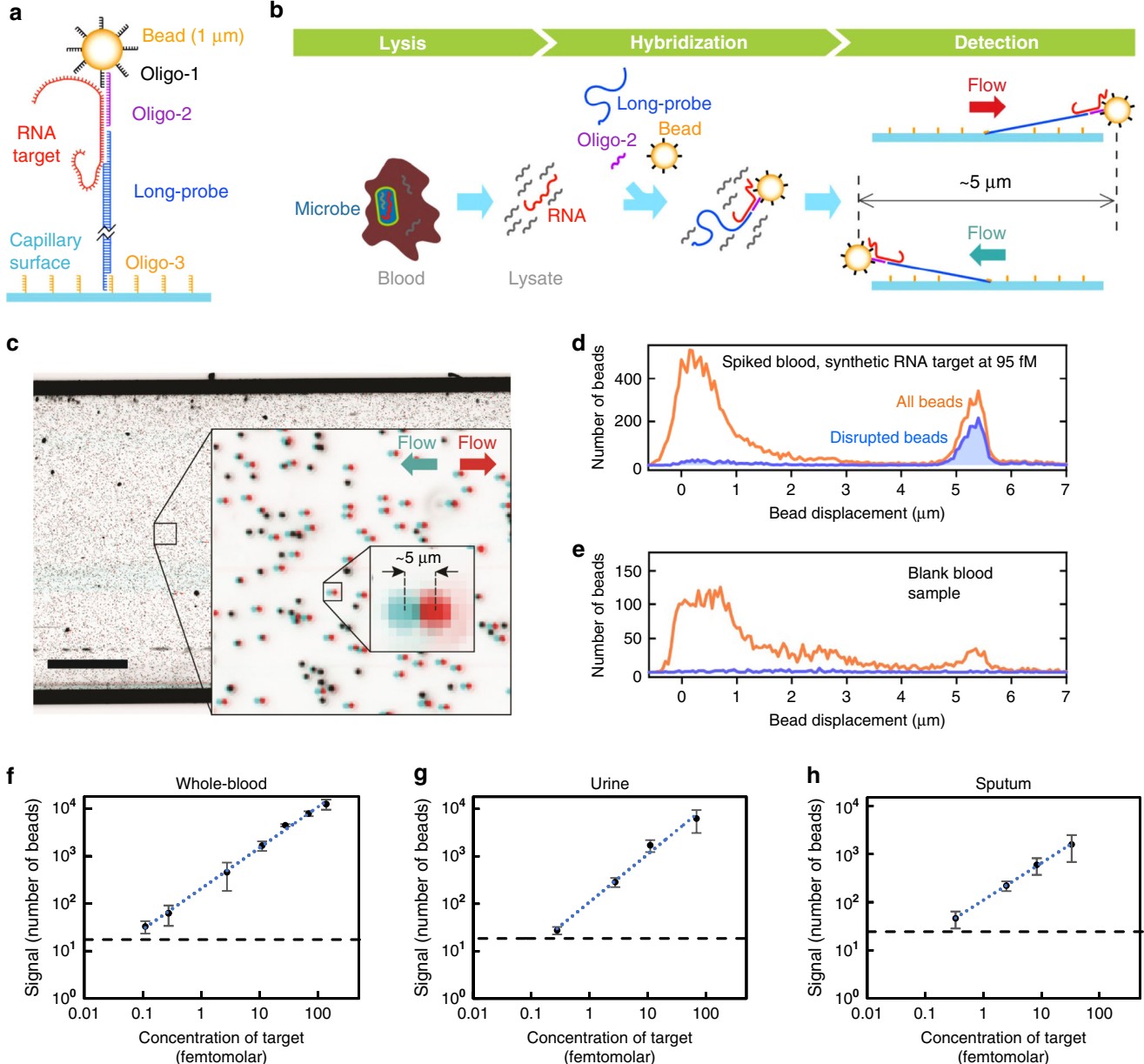

**Fig. 1 SMOLT technology design and validation. a** SMOLT biosensor design. Superparamagnetic beads are functionalized with a DNA Oligonucleotide (Oligo-1). Oligo-2 has a section complementary to Oligo-1, and a section complementary to the target. The Long Probe (>3 kb pairs) is mostly double-stranded with single-stranded ends. One end is complementary to the target while the other is complementary to an oligonucleotide chemically linked to the capillary (Oligo-3). **b** The SMOLT assay protocol consists of body fluid lysis, incubation with probes and beads, and capillary detection where beads are imaged with flow in opposite directions. **c** Superposition of two false-color capillary images exemplify SMOLT bead displacement determination. The red image was obtained with flow in one direction and the blue obtained with flow in the opposite direction. The displacement of a bead is its change of position between the two images. Non-displaced beads appear black. Similar images were obtained in each bead displacement analysis of this study, $n >$ 200. Scale bar, 1 mm. **d**, **e** Histograms of bead displacement. The orange line histograms include the displacement of all the beads in the imaged area. The blue line histograms only include the displacement of disrupted beads (see text). Human whole blood **d** spiked with target, **e** not spiked (blank). Detection of synthetic RNA target spiked into human **f** whole blood, **g** urine, **h** sputum. The cutoffs (black dashed line) were defined as the mean plus two times the standard deviation (SD) of blank sample signals in the corresponding body fluid. The dotted blue line in each panel is a linear function fitted to the data. Data are represented as mean ± SD. Number of experiments in whole blood were $n = $ 5, 4, 6, 3, 3, 3, 3 for concentrations 0.11, 0.28, 2.77, 11.07, 27.68, 69.19, 138.38 fM, respectively. Number of experiments in urine were $n = $ 4, 3, 3, 3 for concentrations 0.27, 2.77, 11.07, 69.19 fM, respectively. Number of experiments in sputum were $n = $ 4, 3, 4, 3 for concentrations 0.33, 2.49, 8.30, 33.21 fM, respectively. Source data are provided as a Source Data file.

sputum (Fig. 1f–h and Supplementary Fig. 4). To our knowledge, sub-femtomolar LODs for detection of nucleic acids in body fluids have been reported only by three methods that do not require enzymatic target amplification. In contrast to SMOLT, these methods require nucleic acid purification and long processing times (6–30 h)[13,17,18]. In the context of viremia detection,

some amplification-based assays detect RNA in whole blood with sub-attomolar LODs[19,20]. Amplification-based assays that target bacterial ribosomal RNA (rRNA) in whole blood are not as sensitive and have LODs comparable to SMOLT's LODs[21,22]. This difference in sensitivity is likely a consequence of the large volume of blood required for bacteria detection, which makes

removal of polymerase inhibitors difficult. The amplification assays that target rRNA in whole blood are complex and require relatively long processing times (5–6 h). These results demonstrate the extremely low LOD of SMOLT for detection of RNA molecules in a variety of body fluids.

**Detection of microbes directly in whole blood.** We next explored the clinical performance of SMOLT by detecting microbes on whole-blood specimens. Microorganisms in the bloodstream can trigger sepsis, a condition that requires immediate treatment[23]. However, the current gold standard for sepsis diagnosis is blood culture, which takes 1–5 days[24]. Here, we show that SMOLT detection of microbial rRNA enables high sensitivity with a turnaround time from sample to results of 1.5 h. We designed species-specific probes that targeted the rRNA of the two most prevalent sepsis-causing fungi, *Candida albicans* and *Candida glabrata*[25], as well as pan-fungal probes that targeted highly conserved regions in fungal rRNA using a local database of microbial and human rRNA sequences (see "Methods" and Supplementary Tables 2, 3). We spiked 5 mL of whole blood from healthy donors with microorganisms to study the performance of each probe pair (Long Probe and Oligo-2 with sequences complementary to the target). The large sample volume was necessary to minimize cell concentration sample-to-sample variability given

that the actual number of cells in the sample follows a Poisson distribution (see "Methods"). After releasing the microbial rRNA through lysis, only a 0.1 mL aliquot from the lysate was analyzed. We studied the detection of *C. albicans* and *C. glabrata* with their species-specific probe pairs and of *Candida parapsilosis* with the pan-fungal probe pair over the range of clinically relevant concentrations (1–100 CFU mL$^{-1}$) and found that the SMOLT signal increased linearly over this range (Fig. 2a–c)[5]. We also found that the *C. albicans* and *C.* glabrata-specific probe pairs were able to detect the corresponding organisms at concentrations as low as 2 CFU mL$^{-1}$ (Fig. 2d, e). The pan-fungal probe pair was able to detect each of the *C. albicans*, *C. glabrata*, and *C. parapsilosis* at 1 CFU mL$^{-1}$ (Fig. 2f). These measurements demonstrate that SMOLT has LODs at least four orders of magnitude lower than the LODs reported by other polymerase-free molecular technologies for detection of microorganisms in blood[13,14]. Furthermore, SMOLT has a turnaround time that is at least threefold faster than polymerase-based methods that have LODs as low as SMOLT[9,10]. The specificity of each probe pair was tested by spiking whole-blood samples at a concentration above the clinical range. In these tests, samples were spiked with 1000 CFU mL$^{-1}$ of *C. albicans*, *C. glabrata*, *C. parapsilosis*, or *Escherichia coli* plus *Staphylococcus aureus*. We found that all probe pairs were highly specific, with no cross-reactivity signal above the corresponding cutoff (Fig. 2d–f).

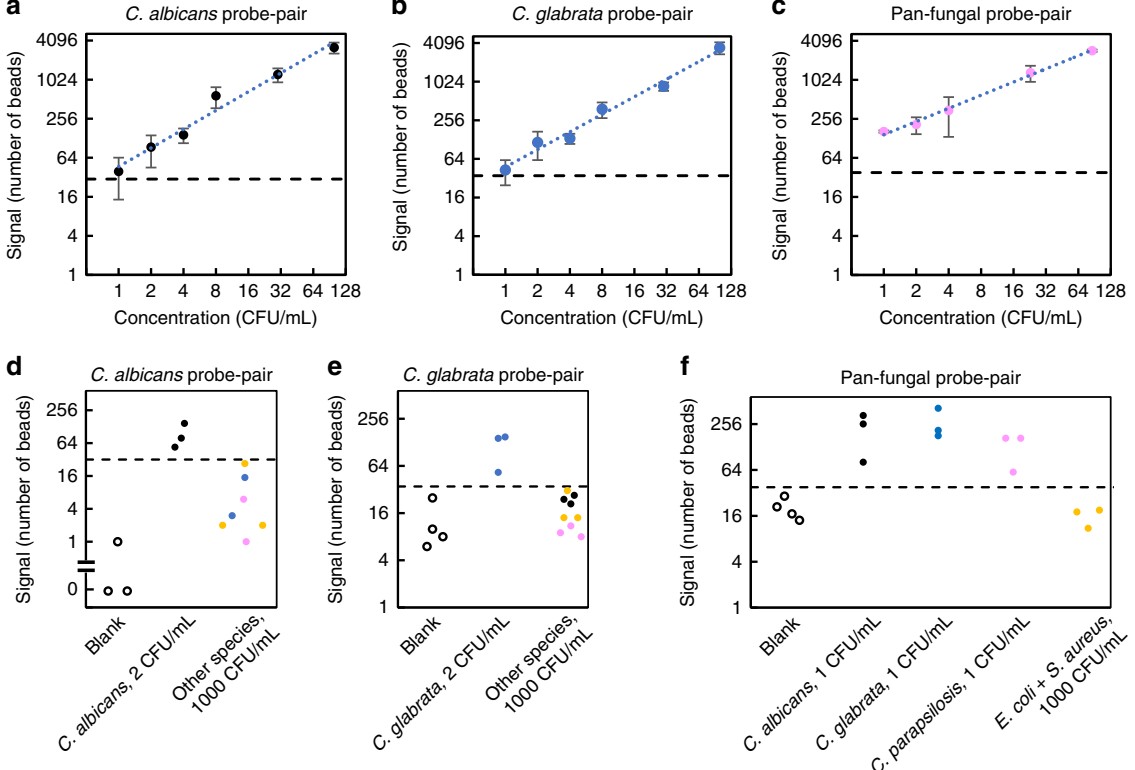

**Fig. 2 SMOLT detection of *Candida* in human whole blood.** Human whole blood samples were spiked with *Candida* cells at indicated concentrations. **a–c** Detection of *Candida* across the range of clinical concentrations. Data plotted on Log$_2$ scale for both axes. The signal increased with increasing concentrations. The dotted blue line in each panel is a linear function fitted to the data. Data are represented as mean ± SD, $n = 3$ independent experiments. **a** *C. albicans* probe pair detecting *C. albicans* ($R^2 = 0.9731$). **b** *C. glabrata* probe pair detecting *C. glabrata* ($R^2 = 0.9831$). **c** Pan-fungal probe pair detecting *C. parapsilosis* ($R^2 = 0.9958$). The cutoffs (black dashed line) were defined above the mean plus two times the SD of the blank samples signal for each probe pair. LOD and cross-reactivity of the **d** *C. albicans* probe pair; **e** *C. glabrata* probe pair; and **f** pan-fungal probe pair. Blank samples (empty circles), *C. albicans* (black circles), *C. glabrata* (blue circles), *C. parapsilosis* (pink circles) and *E. coli* plus *S. aureus* (yellow circles). The LOD was defined as the concentration for which three measurements were above cutoff with no measurement below cutoff. At each LOD concentration, the signal was significantly different from the blank signal, each with $p < 0.05$. Unpaired two-tailed Student's $t$ test was used for comparing two concentrations and $p$ values are **d** $p = 0.028$, **e** $p = 0.011$, **f** $p = 0.022$ for *C. albicans*, $p = 0.0097$ for *C. glabrata*, $p = 0.014$ for *C. parapsilosis* using pan-fungal probe pair. Samples spiked with *E. coli* plus *S. aureus* had 1000 CFU mL$^{-1}$ of each species. Source data are provided as a Source Data file.

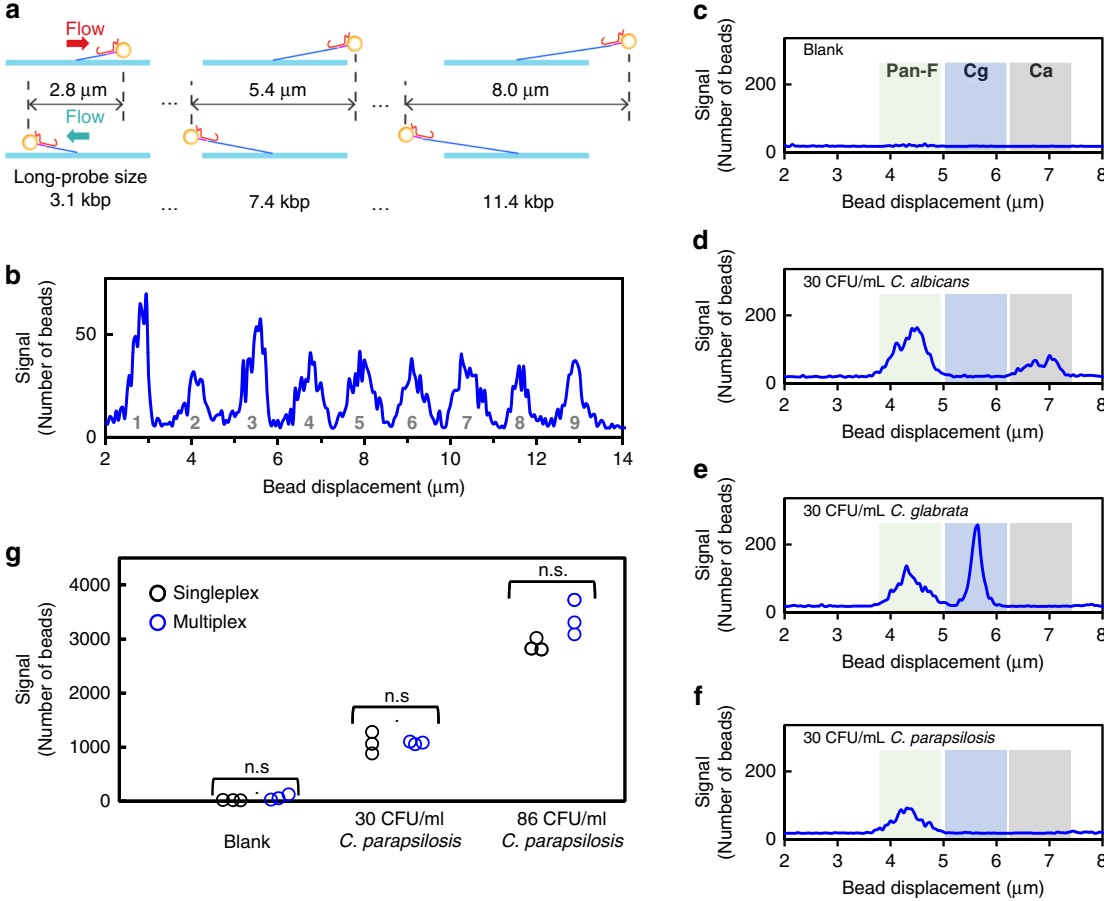

**Fig. 3 SMOLT multiplex detection. a** Strategy for detection of multiple targets in a single reaction and single capillary. Each target is detected with a Long Probe of a different length. Beads tethered by a Long-Probe move a distance that is correlated to the Long-Probe length and can be differentiated from beads tethered with probes of a different length based on displacement. **b** Demonstration of SMOLT multiplexing by detecting rRNA targets using nine Long-Probes of different lengths that generate nine unique peaks in the histogram of bead displacement. Peaks 1–9 correspond to Long-Probes with the following lengths: 3.1, 5.2, 7.4, 9.4, 11.4, 13.4, 15.4, 17.4, and 19.4 kb. **c–f** Histograms of bead displacement, including only disrupted beads, for multiplex detection of *Candida* spiked into human whole blood. Three probe pairs simultaneously detected three target sequences. A representative histogram of three independent experiments is shown. **c** Blank sample, **d** 30 CFU mL$^{-1}$ *C. albicans* only, **e** 30 CFU mL$^{-1}$ *C. glabrata* only, and **f** 30 CFU mL$^{-1}$ *C. parapsilosis* only. Pan-F pan-fungal probe pair peak (green window); Cg *C. glabrata* probe pair peak (blue window); Ca *C. albicans* probe pair peak (gray window). The length of the Pan-F, Cg, and Ca Long Probes are 5.4, 7.4, and 9.4 kb, respectively. As expected, *C. albicans* generated the pan-fungal peak and *C. albicans*-specific peak. Similarly, *C. glabrata* generated the pan-fungal peak and *C. glabrata*-specific peak. In contrast, *C. parapsilosis* only generated the pan-fungal peak. **g** Signal generated from multiplex detection versus singleplex detection of either blank, 30 CFU mL$^{-1}$ or 86 CFU mL$^{-1}$ of *C. parapsilosis*. Analysis using unpaired two-tailed *Student's t*-test showed that signal generated from multiplex and singleplex detection are indistinguishable for each comparison with *p* value > 0.1 (n.s. not significant).

To further demonstrate the broad applicability of the SMOLT assay, we developed and tested probes for detection of two of the most common bacteria found in sepsis patients, *S. aureus* and *Pseudomonas aeruginosa* and observed a LOD of 3 CFU mL$^{-1}$ for both species (Supplementary Fig. 5).These results demonstrate that the SMOLT assay has sensitivity, specificity and performs across a dynamic range consistent with the requirement of a sepsis diagnostic assay.

**Multiplex detection of *Candida* species directly in whole blood.** Multiplex detection is highly desirable when information regarding multiple targets is needed in a diagnostic assay. The use of Long Probes enables a straight-forward approach for multiplex detection. By using Long Probes of different lengths, multiple targets can be detected in a single capillary (Fig. 3a). Beads tethered by a Long Probe of a specific length move the same distance and generate a distinguishable peak in the bead displacement histogram (Fig. 3b). In this approach, the capillary is homogenously functionalized with

the same single-stranded DNA molecule (Oligo-3) without the need of an array. To demonstrate this SMOLT capability in a clinically relevant assay, we developed a multiplex assay with the two *Candida*-specific probe pair and the pan-fungal probe pair. Figure 3c–f shows multiplex detection with the three probe pairs. Three distinct peaks with well-defined boundaries were formed, allowing accurate identification of species. As expected, both *C. albicans* and *C. glabrata* generated two peaks: a species-specific and a pan-fungal peak. In contrast, *C. parapsilosis* generated a single pan-fungal peak, demonstrating detection without cross-reactivity from the species-specific probes. Multiplex and singleplex detection produced signals of the same magnitude (Fig. 3g).

## Discussion
SMOLT detection of pathogens in blood is at least threefold faster than polymerase-based methods with comparable LODs[9,10]. The SMOLT turnaround time can be further reduced by optimizing the hybridization step which is the most time-consuming step of

the protocol with a total duration of 45 min. The hybridization step includes DNA probes hybridization to the target molecule to form a complex and bead binding to that complex. The duration of this step was optimized in this study to achieve reaction completion at relatively low DNA probes and bead concentration (see "Methods"). Higher DNA probe and bead concentrations can enable shorter incubation times. The success of this optimization will depend on the incidence of the higher concentrations on the background signal and the capacity of the disruption step to discriminate additional beads non-specifically bound to the Long Probe.

The SMOLT imaging system is robust, compact, and cost-effective. The imaging system enables measuring the displacement of thousands of micron-size beads with high precision over a field of view of several millimeters square (Fig. 1c and Supplementary Fig. 2). The system's hardware has a total cost of less than US $3000 and consists of light emitting diodes (LEDs) that generate dark field illumination, a telecentric lens with 1× magnification and low numerical aperture, and a 14 megapixel digital camera (see Supplementary Fig. 1 and "Methods"). Highly precise measurement of bead displacement is possible because the image of a bead is distributed over multiple pixels and, therefore, the bead center can be calculated as the average position of those pixels. SMOLT utilizes standard image processing algorithms to obtain the displacement of all the beads in the field of view (see "Methods").

The information of bead displacement and disruption generates a digital signal based on single-molecule counts. In SMOLT, a single target molecule can mediate the tethering of a bead. The SMOLT signal is generated by counting these tethered beads, which are identified because they are displaced a certain distance and are disrupted. Other digital assays that count single molecules have been developed[26,27]. These systems separate the sample in thousands of droplets or wells and have demonstrated a superior performance over traditional analog approaches but require more complex hardware. Significant efforts are underway to simplify these systems in order to expand their clinical use[28–30].

Many polymerase-free and PCR-based assays have been developed with a variety of performance capabilities (Table 1). In comparison to other assays, the advantages of SMOLT are its low LOD for microbe detection in whole blood, rapid turnaround time, and low instrumentation cost. Other assays, like the nCounter, have higher multiplexing capacity than SMOLT. However, SMOLT can detect more targets in a single reaction than other highly sensitive microbe detection techniques and this is an important advantage for the diagnosis of infections where multiple pathogens need to be identified.

In conclusion, the SMOLT method maximizes assay performance by lysing and detecting directly in body fluids and by tethering beads with single molecules when target molecules are present. SMOLT achieves rapid detection with high sensitivity and specificity without enzymatic amplification nor fluorescence detection in body fluids without pre-treatment. SMOLT is ideal for blood-based assays because it can analyze whole blood without the complex sample preparations schemes that are required in polymerase-based methodologies to remove polymerase inhibitors. In the SMOLT assay, over 6000 non-specifically attached beads are typically discriminated (Supplementary Table 1) and over 10,000 beads specifically tethered to the capillary surface can be identified (Fig. 1f). Synthetic RNA detection in whole blood has a large linear dynamic range of over three orders of magnitude and the signal for microbe detection in whole blood is linearly correlated with pathogen concentration at the clinically relevant range between 1 and 100 CFU mL$^{-1}$. In addition, the simplicity of the assay, including reagents, biochemical process, multiplexing mechanism, and instrumentation reduce costs as

**Table 1 Key properties of representative polymerase-free and PCR-based technologies.**

| | Polymerase-free | | | | PCR-based | |
|---|---|---|---|---|---|---|
| | SMOLT | Simoa[13,27,36] | nCounter[17] | Electrochemical sensor[14] | Xpert[37–39] | T2Dx[9,10] |
| LOD for single-stranded nucleic acid (fM) | 0.1 (whole blood) | 0.1 (buffer) | 0.1–0.5 (cell culture) | - | 7·10$^{-7}$ (plasma) | - |
| LOD for microbe in whole blood (CFU mL$^{-1}$) | 1–3 | 4·10$^4$ | - | 10$^7$ | 10 | 1–11 |
| Turnaround time (h) | 1.5 | 5.5 | >20 | 1 | 1.5–2 | 4–8 |
| Multiplexing (number of targets) | >9 | 10 | >500 | 16 | 3 | 5 |
| Instrumentation cost | Low | High | High | Low | High | High |

well as turnaround time and makes SMOLT an attractive technology for the distribution of molecular diagnostics closer to patients and physicians. These advantages suggest that SMOLT has the essential features needed for the development of molecular assays that could significantly improve patient diagnosis and clinical care.

## Methods

**Cell culture.** *C. albicans* (catalog no. 90028), *C. glabrata* (catalog no. 90030), *C. parapsilosis* (catalog no. 90018) and *S. aureus* (catalog no. 29213) were purchased from ATCC. *E. coli* (bank no. 0077) were provided by the Antibiotic Resistant Isolate Bank of Centers for Disease Control and Prevention (CDC). The three *Candida* species were grown on YM agar/broth (catalog no. 271120, BD Biosciences). *S. aureus* was grown on Tryptic Soy agar/broth (catalog no. 211825, BD Biosciences). *E coli* was grown on LB agar/broth (1% w/v Tryptone, 0.5% w/v Yeast extract, 0.5% w/v NaCl, 0.1% v/v 1 N NaOH). For spiking experiments, a single colony of bacteria or yeast was inoculated into 2 mL of the respective broth and incubated overnight in a shaker incubator at 250 rpm at 37 °C (for bacteria) or 30 °C (for yeast). After overnight culturing, cells were spun down and resuspend in 10 mM Tris (pH 8.0). Optical density of the cells was then measured using NanoDrop 2000c at the wavelength of 600 nm. As described in the next section, the measured optical density (OD) of the cells was used to determine the concentration of the cells (i.e., CFU mL$^{-1}$) using a conversion factor obtained for each species.

**Cell concentration (CFU mL$^{-1}$) versus OD.** The optical density of resuspended cells from an overnight culture was measured and then serial dilutions of cell suspension were made. Ten to 50 μL of the last two dilutions were spread onto respective agar plates with 3 mm glass beads. Number of colonies formed on the agar plates were counted after incubation at 37 °C overnight for bacteria or at room temperature for 2 days for yeast. At least two independent plating experiments were conducted for each species used in this study. Number of colonies (i.e., colony forming unit or CFU) counted were used to determine the relationship between cell concentration (i.e., CFU mL$^{-1}$) and OD For *C. albicans*, OD of 1 corresponds to $5.77 \times 10^6$ CFU mL$^{-1}$; for *C. glabrata*, OD of 1 corresponds to $1.17 \times 10^7$ CFU mL$^{-1}$; for *C. parapsilosis*, OD of 1 corresponds to $9.88 \times 10^6$ CFU mL$^{-1}$; for *E. coli*, OD of 1 corresponds to $1.00 \times 10^8$ CFU mL$^{-1}$; for *S. aureus*, OD of 1 corresponds to $6.28 \times 10^8$ CFU mL$^{-1}$.

**Synthetic oligonucleotides.** Synthetic oligonucleotides used in this study were custom synthesized by Integrated DNA Technologies. The RNA oligonucleotide target used in Fig. 1f–h was 57 nucleotides long. The sequences of the probe pair used to detect this synthetic RNA oligonucleotide are listed in Supplementary Table 2. Oligo-1, Oligo-2, and Oligo-3 were DNA oligonucleotides. Oligo-1, used to functionalize beads, had an amine group at the 5′ end and its sequence was a tandem repeat 30-nucleotide long. The sequence of Oligo-2 consisted of two parts: a 5′ segment that was complementary to the corresponding target sequence, and a 3′ segment that was complementary to Oligo-1. Oligo-3, used to functionalize the glass capillary, had an amine group at the 5′ end and its sequence was a random sequence of 36 nucleotide.

**Functionalization/conjugation of beads and capillary.** Dynabeads™ MyOne™ Carboxylic Acid (catalog no. 65011, Invitrogen) one-micron magnetic beads were covalently linked to the amine modified Oligo-1 following the manufacturer's instructions. Rectangle hollow glass capillaries (50 mm × 4 mm × 0.2 mm, $L \times W \times D$) were purchased from VitroCom (catalog no. 3524-050). Glass capillaries were functionalized with 5′-amino modified Oligo-3 via aldehyde-terminated silane. Bare glass capillaries were rinsed with ethanol solution and dried thoroughly with nitrogen gas. Surface activation of the capillaries was done by oxygen plasma treatment (Plasma Etch, Inc.). The activated capillaries were silanized with triethoxysilylundecanal (catalog no. SIT8194.0, Gelest) using chemical vapor-phase deposition[31]. After the silanization, capillaries were filled with the Oligo-3 solution (25 μM Oligo-3 in 100 mM Phosphate Buffer, pH 7.26, and 50 mM NaCNBH$_3$ (catalog no. 296945, MilliporeSigma) for 20 h at room temperature. Subsequently, non-functionalized aldehyde groups were blocked with 0.5 M ethanolamine solution (pH 8.5, catalog no. E9508, MilliporeSigma) and 50 mM NaCNBH$_3$ for 1 h at room temperature. At the final steps, Oligo-3-functionalized capillaries were rinsed twice and dried with nitrogen gas. Functionalized capillaries were stored in a desiccator until use.

**Generation of Long Probes.** A Long Probe was generated through three steps: linearization, ligation, and purification. First, a maxi-prep (catalog no. K210007, PureLink® HiPure Plasmid) purified DNA plasmid was subjected to restriction enzyme digestion to generate a linearized plasmid. The linearized plasmid had two distinct four base overhangs at its two ends. The length of the DNA plasmids used in this study ranged from 3.1 to 19.4 kb. In the second step, the two ends of the linearized plasmid were ligated with two different DNA duplexes generated by hybridizing synthetic DNA oligonucleotides. Each DNA duplex consisted of three

parts: (i) a four base 5′ overhang, complementary to either overhang of the linearized plasmid; (ii) a 25 bp or 27 bp double-stranded DNA region, which facilitated ligation by T4 DNA ligase; and (iii) a 3′ single-stranded DNA flap that was either complementary to Oligo-3 or to the target sequence. Duplexes were ligated to the linearized plasmid using T4 DNA ligase (catalog no. M0202, New England Biolabs) at room temperature for 2–16 h. The ligated DNA product was referred as the Long Probe. In the third step, the Long Probe was purified using the Select-a-Size DNA Clean & Concentrator MagBead Kit (catalog no. D4085, Zymo Research) to remove enzymes, salt and un-ligated DNA duplexes.

**Human body fluid samples.** Human whole blood from healthy donors were purchased from ZenBio (catalog no. SER-WB, EDTA as anticoagulant). Sputum samples were remnants of samples obtained for clinical purposes. The remnants were no longer needed and were going to be discarded. These samples were de-identified and provided by the Johns Hopkins Hospital Medical Mycobacteriology Laboratory. Human urine samples were self-provided by a scientist conducting the experiments with informed consent.

**Considerations regarding blood sample size.** We spiked 5 mL of whole-blood samples from healthy donors with *Candida* species in this study. The 5 mL volume was chosen because microorganism concentrations in blood are as low as 1 CFU mL$^{-1}$. Thus, it is critical to process a sufficiently large sample to ensure that microorganisms will be present. The number of microorganisms $k$ in a sample follows a Poisson distribution, $(k) = e^{-C \cdot V} C \cdot V / k!$, where $C$ is the concentration of microorganisms in the patient's bloodstream and $V$ is the sample volume. If $C = 1$ CFU mL$^{-1}$, $V$ must be at least 5 mL for the probability of obtaining no cells in the sample to be less than 1%.

**SMOLT assay procedure.** SMOLT assay consists of three steps: (1) lysis, (2) hybridization, and (3) detection. Details of each steps are described below.

During lysis, all three human body fluids (i.e., blood, sputum, urine) were diluted 2.5-fold volume in distilled water and lithium dodecyl sulfate (LiDS, catalog no. 50-997-906, ThermoFisher or catalog no. L9781, MilliporeSigma) to a final of 8% LiDS. In RNA oligonucleotide detection experiments (i.e., data presented in Fig. 1f–h and Supplementary Fig. 4), diluted human body fluids were lysed at 95–100 °C for 4 min in a sand bath inside an oven. Lysed body fluids were then spiked with RNA molecules. In the *Candida* detection experiments, diluted and spiked whole-blood samples were lysed inside a capped glass vial (catalog no. 14-962-26F, ThermoFisher) which was introduced into a boiling water bath for 4 min. Note that the reported RNA oligonucleotide concentrations and microbial concentrations are concentrations in the body fluid before dilution.

During the hybridization step, 100 or 300 μL of the lysate was mixed with 20 or 40 μL, respectively, of reagents to final concentrations of 250 mM LiCl, 1 or 3 nM Long Probe(s) and 2 or 8 nM Oligo-2(s). This mixture of lysate and DNA probes was incubated in a heat block at 65 °C for 30 min for target-probe hybridization. Then, Oligo-1 functionalized beads were added to the lysate-probe mixture at the concentration of $2 \cdot 10^8$ mL$^{-1}$; the lysate-probe-bead mixture was incubated at 60 °C for 15 min in a rotating oven (catalog no. SI-1400, Scientific Industries). The sample-probe-bead mixture was then diluted with washing buffer (700 mM LiCl, 0.2% Tween 20, 10 mM Tris-HCl pH 8.0, 10 mM EDTA). Next, a magnetic stand (Catalog no: 12321D, ThermoFisher) was used to attract the magnetic beads to the side of the tube and the supernatant was removed. Beads were then resuspended in 50 μL of washing buffer. Last, salmon sperm DNA (catalog no. D7656, MilliporeSigma) was added to the 50 μL hybridized sample to a final concentration of 1.1 mg mL$^{-1}$.

SMOLT detection was performed on the custom-built SMOLT imaging stage (illustrated in Supplementary Fig. 1 and further described in the next paragraph). The 50-μL hybridized sample was flown into the functionalized capillary and let stand for 20 min to allow for bead sedimentation and Long Probe hybridization to Oligo-3 on the capillary surface. Unbound molecules and beads were removed by solution flow. Ten images were acquired with solution flow in one direction (forward images) and another ten images were acquired once the flow was reversed (reverse image). An averaged forward image was obtained by computing the average of the ten forward images. Same process was used to obtain an average reverse image. The position of each bead in the averaged forward and reverse images was defined as the centroid of each bead image. The centroid was found by converting the image into a binary image using a threshold at 40% the maximum intensity and then finding the centroid of the bead pixels. The position of each bead in the averaged images was used to determine the displacement of each bead in the field-of-view (see Fig. 1c). This methodology generated precise measurements of bead displacement with enough resolution to resolve the displacement of beads when they were as close as one micron apart (see Fig. 3a). Bead displacements were summarized in the form of a histogram (example in Fig. 1d, orange traces). After bead displacements were determined, the disruption step was conducted. When detecting synthetic RNA oligonucleotides, 3 mg mL$^{-1}$ RNase A (catalog no. 12091039, ThermoFisher) was flowed into the capillary to detach signal beads. When detecting *Candida* species, 1 μM strand-displacement DNA oligonucleotides was flown into the capillary to remove target-linked beads (sequences in Supplementary Table 2). During the continuous flow of either RNase A or strand-

displacement oligonucleotides, images were taken every minute for 3 or 5 min and beads that left their original location were identified (example in Supplementary Fig. 3c). The original displacement of beads that were removed during the disruption step was also summarized in the form of a histogram (examples in Fig. 1d, e, blue line). The signal in Figs. 1f–h, 2a-f and Supplementary Fig. 4a–c was the number of beads that were displaced a distance between 5 and 5.8 µm and left the capillary during the disruption step. The signal in Fig. 3g was the number of beads that were displaced a distance between 4 and 4.8 µm and left the capillary during the disruption step.

**SMOLT-imaging stage description**. To acquire capillary images and control the flow of liquids inside the capillary, we built an imaging stage that was connected to a syringe pump and a computer (illustrated in Supplementary Fig. 1). The SMOLT imaging stage was built with optical breadboards (THORLABS) to host: (1) a 14-megapixel CMOS camera (two types were used: model no. G1TD14C, JPLY Electronic Technology Co., and catalog no. acA400-10uc, BASLER); (2) a ×1 magnification and 0.082 NA telecentric lens (catalog no. 58430, Edmund optics), (3) a custom made capillary holder; and (4) a LED ring (catalog no. SAW34072, Polytec). The capillary was secured on the capillary holder via four screws. One end of the capillary tube was connected to a syringe pump (catalog no. NE-1000, New Era Pump Systems) through plastic tubing, while the other end was connected to a 5-mL pipette tip through a short plastic tubing. The syringe pump was used to control the speed and direction of the flow inside the capillary. The 5-mL pipette tip was used to introduce reagents into the capillary, including the hybridized sample, the washing buffer and the solution with disruptor molecules. The LED ring was positioned over the capillary to illuminate the beads during imaging. The camera was positioned below the capillary and connected to a computer to acquire capillary images.

**SMOLT probe design**. A list of 222 sepsis-causing bacteria and fungi were compiled based on commercial identification panels[32,33]. Ribosomal large subunit (LSU) and small subunit (SSU) sequences were downloaded from the SILVA ribosomal RNA gene database (https://www.arb-silva.de/)[34]. Human rRNA sequences were downloaded from NCBI (LSU rRNA: accession NR_003287.2; SSU rRNA: accession NR_003286.2; pre-ribosomal rRNA: accession NR_046235.1). We searched for species-specific and pan-fungal sequences. We found six $C.$ $albicans$-specific and five $C.$ $glabrata$-specific regions in the LSU. The rRNA sequence of other fungi in the database had at least one mismatch with respect to these species-specific sequences. Similarly, we found five pan-fungal regions in the SSU. Each pan-fungal region had at least five mismatches to human and bacterial species. Then, we tested these sequences as SMOLT probes (i.e., Long Probe and Oligo-2 in Fig. 1a). We found significant differences in the sensitivity of these probes likely reflecting differences in rRNA accessibility[35]. For species-specific and pan-fungal detection, we selected the pair of probes that gave the largest signal. Final SMOLT probes for $C.$ $albicans$-specific detection, $C.$ $glabrata$-specific detection, and pan-fungal detection are listed in Supplementary Table 2. Examples of sequence alignment of two fungal rRNA sequences targeted by SMOLT probes are presented in Supplementary Table 3.

**Reporting summary**. Further information on research design is available in the Nature Research Reporting Summary linked to this article.

## Data availability
The authors declare that the data supporting the findings of this study are available within the paper and its supplementary information file. Source data are provided with this paper.

## Code availability
Octave (version 5.1.0) was used to acquire bead images and to analyze them. Octave includes image processing functions that were used to conduct each of the image processing steps described in methods (SMOLT assay procedure). Data analysis was conducted in Microsoft Excel (version 16.0.13001.20266).

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

## Acknowledgements

The authors would like to thank Gregory Bowman and Karen Carroll at Johns Hopkins University for discussions that helped to improve the manuscript significantly and Nikki Parrish at Johns Hopkins Hospital for providing sputum samples. Research reported in this publication was supported by the National Institute of Allergy and Infectious Diseases and by the National Institute on Aging of the National Institutes of Health under Awards R44AI124871, R44AI122527 and R43AG056208. The content is solely the responsibility of the authors and does not necessarily represent the official views of the National Institutes of Health.

## Author contributions

A.C. conceived SMOLT including the use of Long Probes and disruption. A.C., W.-C.C. and T.H. designed the experiments. A.C. and W.C. wrote the manuscript. W.-C.C., T.H., and M.Z. performed experiments and statistical analysis. M.Z. prepared functionalized capillaries and beads. G.R., S.M., H.Z., and J.R. performed experiments. Z.X. designed the optical system and performed experiments. R.E.R. provided guidance on assay development as well as clinical applicability and edited the manuscript.

## Competing interests

All authors except R.E.R. are employees of Scanogen Inc. or were employees at the time experiments were conducted and have an ownership position or ownership option position in the company. R.E.R. has no competing interest.
