## [Peer Review File · Nature Communications]

REVIEWER COMMENTS

Reviewer #1 (Remarks to the Author):

The authors present a clever non-PCR technique for the detection of sparse nucleic acids in clinical samples. By imaging the displacement of tethered microbeads, the authors can achieve digital detection, resolving individual nucleic acid binding events. By using this technique, the authors achieve an LOD of about 2 CFU / mL. I recommend this article for publication if the following issues can be addressed:

1. Using synthetic RNA, the authors achieve an LOD of ~ 0.1 fM. The authors do a good job of comparing their system's ultimate LOD for pathogens (CFU/mL) to competing technologies, but do not discuss their LOD for RNA in the context of PCR or ddPCR. This needs to be addressed.
2. The multiplexed detection is particularly attractive. However, it is not clear how to connect the results shown in Fig. 3 to their probe design. I recommend drawing schematics, like in Fig. 1a, which more clearly explain the experimental design to achieve these multiplexed results.
3. The total number of beads evaluated and the dynamic range of the assay are not clearly reported.
4. The use of bead displacement and disruption to decrease background signal is quite clever. However, a more rigorous description of their algorithm to classify positive versus negative beads should be reported. Also, a blinded validation study would greatly strengthen this report, to avoid any possible issues of overfitting and data leakage.
5. More discussion should be given on the required optical readout hardware. The need to resolve very small displacements over a large field of view is unique, and some ideas on how this can be translated from an optical breadboard prototype to a working device in a biomedical / clinical laboratory is necessary for this paper.
6. More consideration of recent techniques to bring digital assays from the laboratory to robust use should be considered more fully and compared to this technology, e.g. Du et al, LOC 2009, Yelleswarapu et al, PNAS 2019, Rane et al, LOC, 2015, ...
7. More discussion and detail is required on the system's turnaround time. What sets that time? What are the required incubation times and how do they scale with the relevant parameters in their system?

Congratulations on the creative and well executed work,

David Issadore

Reviewer #2 (Remarks to the Author):

In this work, the authors have developed a novel technique for rapid identification of potential pathogens from various human body fluids. This technique depends on hybridization of pathogen RNA to a capillary and bead which can then be measured with a fairly simple microscope setup. The objective of this method would be to much more rapidly identify pathogens causing sepsis.

The manuscript is succinct and well-written; the methodology is explained well, and the statistical analysis seems appropriate. The sensitivity of the method is outstanding, as is the rapid turnaround time. This would potentially be a very useful addition to the diagnostic armamentarium.

My comments are as follows. First, in the experiments for Figure 1 using various body fluids (sputum, blood, urine), the authors lyse the fluids PRIOR to addition of the spiked in RNA for detection. This method does allow them to "prove" that they can measure the DNA in various different processed fluids. However, it would be more informative if they were able to spike the RNA in PRIOR to the 5 minute lysis step (which involves heat in a detergent). There have been reports of so-called HERO heat-resistant proteins (Tsuboyama 2020 PLOS biology) that could conceivably complicate detection of the RNA; in addition, there has been one recent report of a set of proteins that become heat resistant when complexed with bacterial nucleic acids (Tetz and Tetz, 2019, Scientific Reports). Thus, demonstrating that the spiked in RNA can be reliably detected in complex fluids subjected to their lysis would be important.

Second, the choice of *Candida* as a pathogen to study is interesting. *Candida* is typically only considered a pathogen when found in the blood, but NOT the sputum or urine, and it tends to only impact patients with several at-risk features. In addition, there are several rapid, non-culture, non-amplification-based assays for *Candida* (such as fungitell, which can be done in a similar time scale to this assay). Bacteria such as *Staph aureus* or *E coli* are at least 5 times more common as causes of blood stream infections, and they can both be pathogens in sputum or urine. Thus, the use of a more common pathogen that is typically found in the fluids they studied for which we lack rapid diagnostics would strengthen the broad applicability of this method. Have the authors attempted this method with bacteria or even viruses (which can be MUCH more difficult to detect)?

Finally, the authors spiked *Candida* in to blood at various concentrations and were able to detect it, including with multiplex capacity. However, the authors are ultimately hoping that this method can be used to detect pathogens from infected hosts; if they were able to obtain biologic samples from either patients or lab animals currently suffering from infections with *Candida* (if that's the pathogen they choose to focus on), this would markedly strengthen their claim that this technique can detect active infections with a rapid turnaround and high sensitivity and specificity, from an organism currently mounting an aggressive inflammatory response to that pathogen. Have the authors considered acquisition of samples from infected hosts to prove that they remain able to detect the pathogen? Were the authors to focus on more common bacterial pathogens, they may be able to identify patients or use model animals with those infections to demonstrate the feasibility of this approach (and also demonstrate the specificity of this method).

Reviewer #3 (Remarks to the Author):

The authors devised a novel amplification free and purification free molecular assay (SMOLT) for detection of microorganisms in body fluids without the need of culturing. The manuscript is not interesting, too general. There are several concerns regarding the novelty and utility of this technique. Thus, the manuscript should be rejected in the highly esteemed journal.

1. There are many studies published for the rapid detection techniques of microorganisms. Thus, more information should be need into the introduction. In the manuscript, "However, many of these methods are time-consuming and have shown insufficient sensitivity for detection of microorganisms, with LODs between 10^3 and 10^7 16 CFU/mL. 11, 13, 14"

2. The proposed primers are too long, how to bypass the primer-dimer problem? In the manuscript, "Beads tethered by a 22 Long-Probe can be discriminated from beads that non-specifically attach to the capillary based on the 23 displacement they exhibit in the presence of liquid flow: Long-Probe tethered beads are displaced a 24 significantly greater distance than non-specifically attached beads"

3. In the manuscript, "During the lysis step, the body fluid is heated to 95°C in the presence of 8 a surfactant at high concentration (8% lithium dodecyl sulfate) to denature enzymes and lyse cells. This 9 highly denaturing environment prevents aggregation and inactivates nucleases that otherwise would 10 degrade target molecules and probes." Why did you choose lithium dodecyl sulfate? Have you tested with other surfactant?

4. Is this technique for RNA only? Any data with double stranded DNA?

5. In case of the microorganism detection, the sample volume is important due to the low level of the microorganism in the samples volume. How to use large sample volume (over 1mL) with this technique?

6. In the manuscript, "To our knowledge, sub-femtomolar LODs for detection of 29 nucleic acids in body fluids have been reported only by three methods that do not require enzymatic 30 target amplification." So far, many highly sensitive detection methods have been developed. What are the advantages and disadvantages of this technique compared to other techniques?

7. In the manuscript, "Furthermore, SMOLT has a turnaround time that is at least 3-fold faster than polymerase-based methods that have LODs as low as SMOLT." I understand the SMOLT is faster than PCR. Please describe the reason why the PCR-free method required for the detection of miroorganisms.

8. How many samples can be detected simultaneously (multiplex)?

9. Need more information regarding the fungal lysis buffer.

10. This technique should be validated with clinical samples, not spike samples.

RESPONSE TO REVIEWER COMMENTS

We would like to thank the reviewers for their time to review our manuscript and their comments that have helped us to improve the manuscript. We made several modification and additions to the text that are highlighted in the new version. We also modified Fig. 3 by adding Fig. 3a and added a new Supplementary Fig. 4 with new experimental data. We thank the reviewers also for their very positive comments about the SMOLT technique and the manuscript: “The authors present a clever non-PCR technique for the detection of sparse nucleic acids in clinical samples”. “The manuscript is succinct and well-written; the methodology is explained well, and the statistical analysis seems appropriate.” “The sensitivity of the method is outstanding, as is the rapid turnaround time. This would potentially be a very useful addition to the diagnostic armamentarium.” “Congratulations on the creative and well executed work.”

Point by point answers to the reviewers’ comments are presented below. Reviewers’ comments are in black font and our responses in blue.

Reviewer #1 (Remarks to the Author):

1. Using synthetic RNA, the authors achieve an LOD of ~ 0.1 fM. The authors do a good job of comparing their system's ultimate LOD for pathogens (CFU/mL) to competing technologies, but do not discuss their LOD for RNA in the context of PCR or ddPCR. This needs to be addressed.

R: We have added the following sentence: “In the context of viremia detection, some amplification-based assays detect RNA in whole-blood with sub-attomolar LODs.^{19,20} Amplification-based assays that target bacterial ribosomal RNA in whole blood are not as sensitive and have LODs comparable to SMOLT’s LODs.^{21, 22} This difference in sensitivity is likely a consequence of the large volume of blood required for bacteria detection which makes removal of polymerase inhibitors difficult. The amplification assays that target ribosomal RNA in whole blood are complex and require relatively long processing times (5-6 h).” (page 6, line 2)

2. The multiplexed detection is particularly attractive. However, it is not clear how to connect the results shown in Fig. 3 to their probe design. I recommend drawing schematics, like in Fig. 1a, which more clearly explain the experimental design to achieve these multiplexed results.

R: We have added Fig. 3a to explain the SMOLT multiplexing strategy more clearly.

3. The total number of beads evaluated and the dynamic range of the assay are not clearly reported.

R: The total number of beads evaluated in a detection experiment is the sum of specifically bound and non-specifically bound beads (orange line histograms of Fig. 1d and Fig. 1e). Supplementary Table 1 lists the total number of beads evaluated for the blank samples in the study. Typically, there are over 6,000

non-specifically bound beads per experiment. The bead numbers reported in Supplementary Table 1 are a good representation of the non-specifically bound beads observed when target molecules are present, except obviously that specifically bound beads are also observed in these experiments. To make this point clear and to report the dynamic range of the assay, we have added the following sentence:

“In the SMOLT assay, over 6,000 non-specifically attached beads are typically discriminated (Supplementary Table 1) and over 10,000 beads specifically tethered to the capillary surface can be identified (Fig. 1f). Synthetic RNA detection in whole blood has a large linear dynamic range of over 3 orders of magnitude and the signal for microbe detection in whole blood is linearly correlated with pathogen concentration at the clinically relevant range between 1 and 100 CFU/mL.” (page 9, line 23)

4. The use of bead displacement and disruption to decrease background signal is quite clever. However, a more rigorous description of their algorithm to classify positive versus negative beads should be reported. Also, a blinded validation study would greatly strengthen this report, to avoid any possible issues of overfitting and data leakage.

R: We have clarified the description of the algorithm to classify positive versus negative beads reported in page 5 line 16 “We define the SMOLT signal as the number of beads that satisfy two conditions: 1) they move a specific distance when the flow in the capillary is reversed and 2) they detach from the capillary during the disruption step. The blue line in **Fig. 1d-e** is a histogram of bead displacement including only the beads that detached from the capillary during the disruption step. The SMOLT signal in **Fig. 1d-e** are the beads in the blue histogram between 5 and 5.8 μm .”

In addition, we added a rigorous description of the algorithm used to determine the assay signal of each figure in the method section “SMOLT assay procedure”.

To further improve the description of the algorithm, we have added the following paragraph: “The information of bead displacement and disruption generates a digital signal based on single molecule counts. In SMOLT, a single target molecule can mediate the tethering of a bead. The SMOLT signal is generated by counting these tethered beads which are identified because they are displaced a certain distance and are disrupted. Other digital assays that count single molecules have been developed.^{26, 27} These systems separate the sample in thousands of droplets or wells and have demonstrated a superior performance over traditional analogue approaches but require more complex hardware. Significant efforts are underway to simplify these systems in order to expand their clinical use.²⁸⁻³⁰” (page 9, line 10)

Regarding the request for blinded studies, we believe the data in its present form is strong. We analyzed 3-5 repeats for each target concentration, and we did not exclude any data. We obtained the data utilizing 3 operators, 8 instruments, and more than 3 human donors for each body fluid.

5. More discussion should be given on the required optical readout hardware. The need to resolve very small displacements over a large field of view is unique, and some ideas on how this can be translated from an optical breadboard prototype to a working device in a biomedical / clinical laboratory is necessary for this paper.

R: We have included the following paragraph: “The SMOLT imaging system is robust, compact, and cost-effective. The imaging system enables measuring the displacement of thousands of micron-size beads with high resolution over a field of view of several millimeters square (Fig. 1c and Supplementary Fig. 1). The system’s hardware has a total cost of less than US\$3,000 and consists of light emitting diodes (LEDs) that generate dark field illumination, a telecentric lens with 1X magnification and low numerical aperture, and a 14 mega pixel digital camera (see Supplementary Fig. 1 and methods). Sub-pixel resolution measurement of bead displacement is possible because the image of a bead is distributed over multiple pixels and, therefore, the bead center can be calculated as the average position of those pixels. SMOLT utilizes standard image processing algorithms to obtain the displacement of all the beads in the field of view (see methods).” (page 9, line 1)

6. More consideration of recent techniques to bring digital assays from the laboratory to robust use should be considered more fully and compared to this technology, e.g. Du et al, LOC 2009, Yelleswarapu et al, PNAS 2019, Rane et al, LOC, 2015, ...

R: Please, see paragraph added to address comment #4.

7. More discussion and detail is required on the system's turnaround time. What sets that time? What are the required incubation times and how do they scale with the relevant parameters in their system?

R: The required time for each step of the process is indicated in page 4, line 6: “The SMOLT assay consists of three steps: lysis (4 minutes), hybridization (45 minutes) and detection (40 minutes) (Fig. 1b)”. The method section titled SMOLT assay procedure provides additional information regarding the duration of each sub-step of the hybridization step: “This mixture of lysate and DNA probes was incubated in a heat block at 65°C for 30 minutes for target-probe hybridization. Then, Oligo-1 functionalized beads were added to the lysate-probe mixture at the concentration of $2 \cdot 10^8$ per mL; the lysate-probe-bead mixture was incubated at 60°C for 15 minutes in a rotating oven (catalog no. SI-1400, Scientific Industries).” The same section provides a detailed description of the detection step including information regarding time: “The 50- μ L hybridized sample was flown into the functionalized capillary and let stand for 20 minutes to allow for bead sedimentation and Long-Probe hybridization to Oligo-3 on the capillary surface.”, and “During the continuous flow of either RNase A or strand-displacement oligonucleotides, images were taken every minute for 3 or 5 minutes and beads that left their original location were identified (example in Supplementary Fig. 2c).

We have added the following discussion about the system’s turnaround time: “SMOLT detection of pathogens in blood is at least 3-fold faster than polymerase-based methods with comparable LODs.^{9, 10} The SMOLT turnaround time can be further reduced by optimizing the hybridization step which is the most time consuming step of the protocol with a total duration of 45 minutes. The hybridization step includes DNA probes hybridization to the target molecule to form a complex and bead binding to that complex. The duration of this step was optimized in this study to achieve reaction completion at relatively low DNA probes and bead concentration (see methods). Higher DNA probe and bead concentrations can enable shorter incubation times. The success of this optimization will depend on the incidence of the

higher concentrations on the background signal and the capacity of the disruption step to discriminate additional beads non-specifically bound to the Long-Probe.” (page 8, line 20)

Reviewer #2 (Remarks to the Author):

1. My comments are as follows. First, in the experiments for Figure 1 using various body fluids (sputum, blood, urine), the authors lyse the fluids PRIOR to addition of the spiked in RNA for detection. This method does allow them to "prove" that they can measure the DNA in various different processed fluids. However, it would be more informative if they were able to spike the RNA in PRIOR to the 5 minute lysis step (which involves heat in a detergent). There have been reports of so-called HERO heat-resistant proteins (Tsuboyama 2020 PLOS biology) that could conceivably complicate detection of the RNA; in addition, there has been one recent report of a set of proteins that become heat resistant when complexed with bacterial nucleic acids (Tetz and Tetz, 2019, Scientific Reports). Thus, demonstrating that the spiked in RNA can be reliably detected in complex fluids subjected to their lysis would be important.

R: Synthetic RNA molecules spiked in unprocessed body fluids behave differently than natural RNA molecules in these fluids. Synthetic RNA is rapidly degraded by RNases unless the fluid is modified to prevent this degradation (Nat Biotechnol. 2015 Jul; 33(7): 730–732.). Natural free circulating RNA molecules are protected by proteins or vesicles (PNAS 2011 Mar 22; 108 (12) 5003-5008). In our hands, high LDS concentration and heat is required before spiking body fluids otherwise the signal is significantly lower than the signal of control experiments in buffer. Unless one could spike an RNA that is protected from degradation, it is not possible to detect spiked RNA in untreated body fluids. Note also that the structure selected to protect the RNA molecule may modify the effect of the lysis process and therefore one would need to test RNA molecules protected with a variety of proteins and vesicles. In the second part of this work, we show that the assay effectively detects pathogen ribosomal RNA that is subjected to our lysis process. Fungi and now bacteria (see next answer) were spiked in whole blood, then the blood was lysed and the microbial RNA detected. We believe the results of our study support the usefulness of the assay for detection of RNA molecules that are protected from degradation in body fluids.

2. Second, the choice of *Candida* as a pathogen to study is interesting. *Candida* is typically only considered a pathogen when found in the blood, but NOT the sputum or urine, and it tends to only impact patients with several at-risk features. In addition, there are several rapid, non-culture, non-amplification-based assays for *Candida* (such as fungitell, which can be done in a similar time scale to this assay). Bacteria such as *Staph aureus* or *E coli* are at least 5 times more common as causes of blood stream infections, and they can both be pathogens in sputum or urine. Thus, the use of a more common pathogen that is typically found in the fluids they studied for which we lack rapid diagnostics would strengthen the broad applicability of this method. Have the authors attempted this method with bacteria or even viruses (which can be MUCH more difficult to detect)?

R: We agree with the reviewer that it is important to show bacteria detection to demonstrate the broad applicability of the assay. We developed and tested probes for detection of *Staphylococcus aureus* and *Pseudomonas aeruginosa* in whole blood and observed LODs comparable to the LODs of fungal species

(Supplementary Fig. 4). We would like to note that the available rapid fungi assays mentioned by the reviewer are not molecular and have low specificity.

3. Finally, the authors spiked *Candida* in to blood at various concentrations and were able to detect it, including with multiplex capacity. However, the authors are ultimately hoping that this method can be used to detect pathogens from infected hosts; if they were able to obtain biologic samples from either patients or lab animals currently suffering from infections with *Candida* (if that's the pathogen they choose to focus on), this would markedly strengthen their claim that this technique can detect active infections with a rapid turnaround and high sensitivity and specificity, from an organism currently mounting an aggressive inflammatory response to that pathogen. Have the authors considered acquisition of samples from infected hosts to prove that they remain able to detect the pathogen? Were the authors to focus on more common bacterial pathogens, they may be able to identify patients or use model animals with those infections to demonstrate the feasibility of this approach (and also demonstrate the specificity of this method).

R: The studies with clinical or animal samples suggested by the reviewer are important, but they are expensive. We will conduct those studies once we secure the necessary funding. In the case of clinical samples, patients suspected of having sepsis are tested with blood culture and all the blood collected is used in the test. Therefore, patients must provide an additional blood sample for our study. Moreover, the rate of blood culture positivity in these patients is very low (<8%) (Clin Lab Med. 2013 Sep;33(3):413-37). Therefore, hundreds of samples are needed to collect a meaningful number of positive samples.

Reviewer #3 (Remarks to the Author):

1. There are many studies published for the rapid detection techniques of microorganisms. Thus, more information should be need into the introduction. In the manuscript, “However, many of these methods are time-consuming and have shown insufficient sensitivity for detection of microorganisms, with LODs between 10^3 and 10^7 CFU/mL.”^{11, 13, 14}

R: The list provided in the manuscript does not aim to be a complete list of the available techniques. We conducted an extensive search and found that the four listed techniques are good examples of the most important techniques for molecular detection of microorganisms without amplification (references 11-14). We believe that the range of LODs we provide is an accurate representation of what have been reported in scientific journals for these techniques.

2. The proposed primers are too long, how to bypass the primer-dimer problem? In the manuscript, “Beads tethered by a Long-Probe can be discriminated from beads that non-specifically attach to the capillary based on the displacement they exhibit in the presence of liquid flow: Long-Probe tethered beads are displaced a significantly greater distance than non-specifically attached beads”

R: Primer-dimer are not a problem in our assay because we do not use target amplification.

3. In the manuscript, “During the lysis step, the body fluid is heated to 95°C in the presence of a surfactant at high concentration (8% lithium dodecyl sulfate) to denature enzymes and lyse cells. This highly denaturing environment prevents aggregation and inactivates nucleases that otherwise would degrade

target molecules and probes.” Why did you choose lithium dodecyl sulfate? Have you tested with other surfactant?

R: LDS and SDS are anionic surfactants that are often used in lysis buffers for nucleic acids isolation. (Noll H, Stutz E. The use of sodium and lithium dodecyl sulfate in nucleic acid isolation. *Methods Enzymol* 1968; XIIB: 129-56). We chose to use LDS because SDS precipitates below 10°C while LDS remains in solution at temperatures close to 0°C. We did not try other surfactants.

4. Is this technique for RNA only? Any data with double stranded DNA?

R: We have used the technique to detect single stranded RNA and single stranded DNA. We have not studied detection of double stranded DNA.

5. In case of the microorganism detection, the sample volume is important due to the low level of the microorganism in the samples volume. How to use large sample volume (over 1mL) with this technique?

R: The technique detects microorganisms in 5 mL of whole blood. Please, see page 6, line 18 and the method section “Considerations regarding blood sample size”.

6. In the manuscript, “To our knowledge, sub-femtomolar LODs for detection of nucleic acids in body fluids have been reported only by three methods that do not require enzymatic target amplification.” So far, many highly sensitive detection methods have been developed. What are the advantages and disadvantages of this technique compared to other techniques?

R: Alternative non-amplified technologies require nucleic acid purification and long processing times (6-30 h). Please, see ref. 13, 17, and 18.

7. In the manuscript, “Furthermore, SMOLT has a turnaround time that is at least 3-fold faster than polymerase-based methods that have LODs as low as SMOLT.” I understand the SMOLT is faster than PCR. Please describe the reason why the PCR-free method required for the detection of microorganisms.

R: SMOLT is at least 3-fold faster, has higher multiplexing capacity and a more straight-forward protocol than amplified methodologies. Polymerase-based methodologies require complex sample preparation steps to remove polymerase inhibitors in certain specimen types, and relatively expensive reagents and instrumentation. These limitations have complicated the development and implementation of PCR-based systems for some applications, such as culture-free sepsis diagnosis.

8. How many samples can be detected simultaneously (multiplex)?

R: The SMOLT capillary imaging stage described in Supplementary Fig. 1 detects one sample at a time. In our lab, scientists routinely use eight of these imaging stages to detect eight samples simultaneously.

9. Need more information regarding the fungal lysis buffer.

R: The lysis buffer composition and lysis procedure are described in the method section “SMOLT assay procedure”.

10. This technique should be validated with clinical samples, not spike samples.

R: Please, see third answer to reviewer #2.

REVIEWERS' COMMENTS:

Reviewer #1 (Remarks to the Author):

Overall the authors did a solid job of responding to the reviews. However, there remains one quite serious issue which I think should be addressed before publication. There remains a lack of rigor and consistency on the description of their optical systems's resolution. The resolution is described as "sub-micron"(in the abstract) and other times as having "micrometer resolution"(Introduction). Despite these claims, there is no quantitative evaluation of the system's resolution given that I can find. This omission is quite serious, in my opinion, given the centrality of sub-pixel resolution to their approach. I recommend strongly the authors measure resolution quantitatively and report it.

Otherwise, I recommend this manuscript for publication.

Reviewer #2 (Remarks to the Author):

I'm satisfied with their responses to my comments. You've addressed them well and appropriately, and I appreciate that testing this modality with either human or animal samples is not trivial... however, it should be the next step that you undertake in pursuing this technology.

Reviewer #3 (Remarks to the Author):

The authors have not fully addressed for the reviewer's comments, especially, number 6, 8, 10.

RESPONSE TO REVIEWERS' COMMENTS

We have addressed the remaining concerns of the reviewers.

Reviewer #1 (Remarks to the Author):

The resolution is described as "sub-micron"(in the abstract) and other times as having "micrometer resolution"(Introduction). Despite these claims, there is no quantitative evaluation of the system's resolution given that I can find. This omission is quite serious, in my opinion, given the centrality of sub-pixel resolution to their approach. I recommend strongly the authors measure resolution quantitatively and report it.

In order to address the indicated consistency and rigor issues, we have changed the term "resolution" for "precision" and added a new supplementary Fig. 2 to support our claim of "sub-micron precision". The capacity of SMOLT to resolve the displacement of probes of multiple lengths is demonstrated in Figures 3b-e.

Reviewer #3 (Remarks to the Author):

The authors have not fully addressed for the reviewer's comments, especially, number 6, 8, 10.

We believe we have answered comments number 8 and 10. Regarding point 6, we added a new paragraph in the discussion section that addresses the limitations and advantages of SMOLT. Because multiple parameters needed to be compared across multiple technologies, we decided to present this comparison with a table (please, see new Table 1 embedded in the text).